# Enhanced Bioactivity of Tailor-Made Glycolipid Enriched Manuka Honey

**DOI:** 10.3390/ijms231912031

**Published:** 2022-10-10

**Authors:** André Delavault, Ahmed E. Zoheir, Delphine Muller, Rebecca Hollenbach, Kersten S. Rabe, Katrin Ochsenreither, Jens Rudat, Christoph Syldatk

**Affiliations:** 1Technical Biology, Institute of Process Engineering in Life Sciences II, Karlsruhe Institute of Technology, 76131 Karlsruhe, Germany; 2Department of Genetics and Cytology, National Research Center (NRC), Cairo 12622, Egypt; 3Molecular Evolution, Institute for Biological Interfaces 1 (IBG-1), Karlsruhe Institute of Technology, 76344 Eggenstein-Leopoldshafen, Germany; 4Technikum Laubholz GmbH, Biotechnologische Konversion, 89143 Blaubeuren, Germany

**Keywords:** glycolipid synthesis, Manuka honey, lipase, esterification, deep eutectic solvent, antimicrobial, stress biosensing

## Abstract

Glycolipids can be synthetized in deep eutectic solvents (DESs) as they possess low water content allowing a reversed lipase activity and thus enables ester formation. Based on this principle, honey can also serve as a media for glycolipid synthesis. Indeed, this supersaturated sugar solution is comparable in terms of physicochemical properties to the sugar-based DESs. Honey-based products being commercially available for therapeutic applications, it appears interesting to enhance its bioactivity. In the current work, we investigate if enriching medical grade honey with in situ enzymatically-synthetized glycolipids can improve the antimicrobial property of the mixture. The tested mixtures are composed of Manuka honey that is enriched with octanoate, decanoate, laurate, and myristate sugar esters, respectively dubbed GOH, GDH, GLH, and GMH. To characterize the bioactivity of those mixtures, first a qualitative screening using an agar well diffusion assay has been performed with methicillin-resistant *Staphylococcus aureus*, *Bacillus subtilis*, *Candida bombicola*, *Escherichia coli,* and *Pseudomonas putida* which confirmed considerably enhanced susceptibility of these micro-organisms to the different glycolipid enriched honey mixtures. Then, a designed biosensor *E. coli* strain that displays a stress reporter system consisting of three stress-specific inducible, red, green, and blue fluorescent proteins which respectively translate to physiological stress, genotoxicity, and cytotoxicity was used. Bioactivity was, therefore, characterized, and a six-fold enhancement of the physiological stress that was caused by GOH compared to regular Manuka honey at a 1.6% (*v*/*v*) concentration was observed. An antibacterial agar well diffusion assay with *E. coli* was performed as well and demonstrated an improved inhibitory potential with GOH upon 20% (*v*/*v*) concentration.

## 1. Introduction

For thousands of years, honey has been applied for its medicinal properties, notably as a wound dressing [1]. From a physico-chemical point of view, it is a viscous, supersaturated sugar solution that is derived from nectar, gathered and modified by the honeybee, and it is possibly one of the few natural deep eutectic solvents (NADES) [2]. This term describes a liquid phase that is obtained by combining hydrogen bond donors (HBDs) with high melting points, which are abundant in nature such as carboxylic acids, sugars, and amides. Their high biodegradability and high recyclability combined with a fairly low water content make them a powerful tool for biocatalysis [3]. As a result, honey is a media that presents interesting properties for chemical and biological purposes.

When pasture honeys are diluted down to 25% of the original volume concentration, they tend to lose their antibacterial activity, most probably due to dilution of the active compounds such as flavonoids, neutralization of pH, and a decrease of osmolarity [4,5]. In a practical case, the dilution of honey, e.g., used as wound gel, can occur when the wound exudates. The “ideal medical honey” would thus have a relatively strong and rapid bactericidal effect compared to standard honeys [6]. In other terms, the antibacterial efficacy would rely on compounds that are effective even upon dilution. With this aim, a good candidate arises: the standardized medical-grade Manuka honey (MH) which exhibits high levels of methylglyoxal (MGO) and shows intrinsic antibacterial activity up to 30% dilution [7]. However, MGO belongs to the α-ketoaldehyde family, a special class of 1,2-dicarbonyl compound, which is highly reactive and as a result non-stable [8,9]. The aim is to strengthen and extend in time the overall antibacterial effect by enriching the media with more stable bioactive compounds that are also effective upon dilution.

Using surfactants for this purpose seems promising; Kwakman et al. tried this principle and obtained convincing results [10]. Indeed, their strategy allowed a more rapid bactericidal effect against antibiotic-resistant pathogens (~2 h) compared to honey alone. However, they tested two surface-active peptides consisting of long amino acid chains (BP2 and LL37, respectively, 19-mer and 37-mer) which were synthetized using solid phase and protection/deprotection technique. Either sequential or convergent, this synthesis method is time costly and expensive, making it less relevant for a larger scale process as honey and lipase formulations are produced on a multi-ton scale at food grade. Parallelly, we established the basics of the DES-mediated enzymatic synthesis of glycolipids, a class of possibly bio-based surfactants. Beyond demonstrating the usefulness of DESs as simple green solvents, we have also shown that sugars could compose a DES, as hydrogen bond donors, in combination with choline chloride and take part, at the same time, in the biocatalysed reaction with fatty acids, forming thus glycolipids such as sugar esters [11]. This DES reaction system, later qualified as “2-in-1”, with DES simultaneously serving as a solvent plus substrate of literally unlimited concentration was, more recently, successfully applied by our group. This time we used honey as a solvent and its components, essentially sugars being fructose and glucose, as substrates [12]. This synthesis relies on the use of the immobilized lipase from *C. antarctica* (iCalB or Novozym 435^®^) for the transesterification reaction between vinyl esters and sugars that are naturally present in honey. Vinyl esters are substrates of choice for enzymatic transesterifications thanks to the vinylic function representing an electrophilic site that offers irreversible and enantioselective acylation of primary hydroxyl groups [13]. The resulting vinyl alcohol as a side product undergoes tautomerization that produces acetaldehyde, a volatile compound that evaporates rapidly due to a low boiling point (20.2 °C) and thus, shifts the reaction equilibrium towards ester formation as depicted in the Figure 1 [14]. This thermodynamic phenomenon favoring the product formation is a practical illustration of Le Chatelier’s principle. Additionally, the great excess of substrates that are provided by honey as sugars should affect the dynamic equilibrium of the reaction in a similar fashion.

The products that were synthetized in honey and agave syrup in our previous work were identified, by means of MS and NMR, as mono- or di-myristate (C14) down to octanoate (C8) esters of fructose and glucose, depending on the vinyl ester that was used and are known to possess antibacterial effects, especially fructose laurate (C12) [15]. But other derivatives such as octanoate, decanoate, laurate, and myristate sugar esters have not yet been tested for bioactivity in combination with honey. According to our previous results, it was shown, for example in the case of a GOH-like mixture (enrichment in octanoate sugar esters), that based on cross peaks of carbohydrate protons with lipid carbonyls in the ^1^H-^13^C HMBC, the glucose moieties were acylated with octanoic acid at the C6 atoms. Thus, with honey as substrate, one clear major carbohydrate system was identified in the sample as glucose based on ^1^H COSY and ^13^C HMBC spectra. Overall, it was determined that the prominently synthesized product in honey was glucose-6-octanoate, thus indicating that the use of iCalB seemed to favor the acylation of glucose at the C6, which was already shown by our group in another context [16]. Whether fructose-octanoate was also a product of the synthesis reaction in honey could however not be conclusively clarified as the molar masses of fructose and glucose are identical (glucose- or fructose-octanoate with a calculated molar mass of 306.168 Da) but the residual formation of sugar-di-octanoates was proven (glucose- or fructose-di-octanoate with 432.272 Da), which is also consistent with our past results

Herein, these reactions are carried out in one of the most efficient honeys against pathogens according to the literature: MH from New-Zealand derived from the Manuka tree (*Leptospermum scoparium*) [17]. Honey as a pasture or wound gel might be a promising alternative strategy to treat surficial wound infections by antibiotic-resistant pathogens [18], since the resistance is increasing worldwide [19] and very few new antibiotics are being developed [20].

In this study, a first qualitative screening with different microorganisms was performed to determine which kinds of bacteria or yeasts were susceptible to honey mixtures that were enriched with different sugar esters. Those microorganisms which are methicillin-resistant *Staphylococcus aureus* (MRSA), *Bacillus subtilis*, *Candida bombicola*, *Escherichia coli,* and *Pseudomonas putida* represent a versatile testing pool with yeast, Gram– and Gram+ microorganisms to determine what could be the targets of such mixtures from a pharmaceutical point of view.

Recently, Zoheir et al. [21] developed a new biosensor system for multiple stress sensing which allows the monitoring of internal and external stresses in the model organism *E. coli* K12 MG1566 cells by using a three-colored fluorescent protein combination. These proteins function throughout three principal stress response mechanisms that are the RpoS pathway (starvation) [22], RpoH (heat-shock) [23], and SOS (DNA-damage) [24], all of which are controlled by three specific promoters. These promoters were selected as (1) P*sulA*, a promoter that was induced during the SOS response that indicates DNA damage (genotoxicity) [25]; (2) P*osmY* from the RpoS regulon that indicates nutrient starvation, osmotic, and other physiological stresses [26]; and (3) P*grpE* which is involved in the heat shock RpoH response and is activated through intracellular accumulation of unfolded proteins (cytotoxicity) [23,27]. This system called RGB-S reporter allows a real-time and simultaneous analysis of the three stress responses in *E. coli* by microplate reading that was equipped with fluorescence detection. Thus, the signals were simultaneously acquired though three orthogonally detectable fluorescent protein variants with red (mRFP1) [28], green (GFPmut3b) [29], and blue (mTagBFP2) [30] colors that were selected to enable the read-out of the aforementioned pathway activations. Once the fluorescent proteins´ downstream of the promoter fused, the assembled RGB-S reporter contained the genetic elements of P*osmY*::mRFP1(physiological stress), P*sulA*::GFPmut3b (genotoxicity), and P*grpE*::mTagBFP2 (cytoxicity). This biosensor system has been used in the current work to characterize the bioactivity of the glycolipid-enriched honey mixtures, thus giving a deeper insight on how a pathogen such as *E. coli* is affected by enriched honey mixtures. Quantitative agar well diffusion experiments also permitted to link measurable inhibition of biofilms on agar plates and multi-stress biosensing. In the end, the most efficient of these mixtures containing octanoate sugar esters (GOH) was compared to the non-enhanced honey to determine upon which concentration glycolipids permit enhancement of the bioactivity on an agar plate.

## 2. Results

### 2.1. Glycolipid Synthesis in Honey

Glycolipids were synthesized in Manuka honey by adding different fatty acid vinyl esters (octanoate (C8, GOH), decanoate (C10, GDH), laurate (C12, GLH), myristate (C14, GMH), and palmitate (C16, GPH)) and immobilized lipase (iCalB). Product formation was qualitatively assessed by thin layer chromatography (TLC). Figure 2 shows similar TLC profiles between GOH, GDH, GLH, GMH, and GPH mixtures. Sugar esters appeared with an average retention factor Rf = 0.65 after dying with anisaldehyde solution as previous work reports from Siebenhaller et al. [12]. Highly polar compounds, such as sugars in this case, have more interaction with the silica-coated layer, therefore, giving a low retention factor (Rf = 0.12).

Unreacted free fatty acids that are hydrophobic present low interaction with the stationary phase and move with the elution front “pushing” them on the very top of the TLC. Glycolipid synthesis was successful with all the applied vinyl fatty acids in the Manuka honey. To firmly support the last claim, a standard that acts as a control was also spotted on the same TLC. This standard (Std) is a commercially available rhamnolipid mixture that was diluted at 10 mg/mL in ethyl acetate which displays glycolipid stain at Rf = 0.68 on normal phase thin layer chromatography. The relative observable intensity of the TLC stains indicates that in each mixture, glycolipids were produced in the same range of concentrations. MHWE serves as a negative control for the glycolipid synthesis since no enzyme was introduced in this condition, only the unreacted vinyl octanoate is present. It shows no characteristic spot of glycolipids affirming that the specific conversion of substrates into glycolipids is due to the enzyme.

### 2.2. Microorganisms Susceptibility Test

This qualitative antimicrobial test was performed with the aim of a pre-selection of glycolipid-enriched honey mixtures using observable inhibition as an orientation. It oriented the work toward the types of microorganisms that show susceptibility to the different glycolipid-enriched MH mixtures. The latter containing palmitate (GPH) was also tested with this assay, but the results showed very poor efficacy and, therefore, was non-relevant for the current work, so was excluded. Table 1 shows that every honey-based mixture that was mentioned above presents bioactivity towards the different microorganisms that were tested. Gram+ and Gram– bacteria being seemingly susceptible, but the yeast species (*C. bombicola*) displayed a very particular susceptibility to the GOH mixture (Figure 3). 

A slight or no effect is observable for MH, MHWE, GDH, GMH, and GLH. GOH showed very clear and large inhibition zones of the biofilm on all the microorganisms that have been tested. *E. coli* and MRSA showed particularly high susceptibility to all the honey mixtures, glycolipid-enriched or not. Thus, highlighting the tendency of every Gram+ microorganism to be susceptible to the MH mixtures. Contrarily, Gram– bacteria and the yeast displayed acute sensibility to mixtures that were enriched with sugar esters.

As *E. coli* was more inhibited by all the glycolipid-enriched honey mixtures than with MH alone, a reporter strain was applied to characterize the type of stress which is exerted by the respective glycolipid-enriched honey mixture.

### 2.3. Whole-Cell Multi-Stress Biosensing and Bactericidal Activity in Broth Dilution

*E. coli* cells that were used as stress biosensors were incubated 24 h with glycolipid-enriched honey mixtures at 37 °C to characterize the apparent bioactivity. In this experiment three different types of stress reporters allow the characterization of the bioactivity. Physiological, cytotoxic, and genotoxic stress were simultaneously detected via fluorescent microplate reading due to three fluorescent proteins giving signals called RFP, BFP, and GFP, respectively. Those signals were normalized to the OD_600_ of *E. coli* to get a signal that is specific to the biomass. Figure 4 displays measures for the different honey mixtures at 1.6% (*v*/*v*) which is the most representative of all the tested concentrations after 20 h of incubation according to Appendix A.

The stress assay showed a 6-fold increase for the physiological stress, a 2.5-fold increase of cytotoxic stress, and no change in the genotoxicity when MH is enriched with octanoate sugar esters (Figure 4B–D). In comparison, other mixtures such as GDH, GLH, and GMH have very low or essentially no effect on *E. coli*´s growth which is reflected by almost negligible stress signals. Moreover, the growth curve displays the rapid effect of GOH compared to the other mixtures with a lag phase of the growing *E. coli* culture starting at early incubation and lasting for 4 h (Figure 4A). Thus, highlighting the fact that, over an extended period, GOH possesses the most efficient and rapid bioactivity of all the mixtures that were tested. MHWE which consisted of the same mixture of MH and vinyl octanoate, except no enzyme was included, showed an obviously smaller effect than the control. Although the differences in bioactivity between MH and MHWE is not as strikingly significant as GOH compared to the rest of the tested conditions considering OD_600_, specific GFP and RFP signals, it is worth noting that the specific BFP signal suggests a higher cytotoxicity of MHWE that could be attributed to the vinylic substrate.

### 2.4. Bactericidal Activity Assay on Agar Plate

*E. coli* was also used for an agar well diffusion assay that was aimed at a quantitative experiment in which the inhibition zone diameters of different concentrations have been measured. Previous results prompted us to compare the bioactivity of MH and GOH in a bactericidal assay upon same dilution range. Concentrations ranging from 0 to 50% (*v*/*v*) with 10% steps were tested. Each condition was repeated throughout three wells. The same dilution ranges were compared across different agar plates. No bioactivity was observed with MH at concentrations that were lower than 40% (*v*/*v*) whereas GOH showed direct inhibition of the microorganism at a concentration of 20% (*v*/*v*). Under this benchmark value, no effect was reported for both mixtures. Diameters of the inhibition zones are reported in Figure 5.

When the concentration in a honey mixture reaches 40% (*v*/*v*), the inhibitory potential of GOH and MH appears to not be significantly different. Although, the GOH mixture obviously exhibits antimicrobial effects at lower concentrations than pure MH.

## 3. Discussion

### 3.1. Impact of the Process on Endogenous and Exogenous Compound in Manuka Honey

The method that was used in the current work for glycolipid production in honey was validated in a previous publication from Siebenhaller et al. [12]. Glycolipids coming from the transesterification of vinyl octanoate (C8), vinyl decanoate (C10), vinyl laurate (C12), and vinyl palmitate (C16) with sugars from honey and agave syrup were successfully identified and characterized using spectrometric methods. Given those previous results, it is assumed that all vinyl esters that were used represent good substrates for Novozym 435^®^ to synthetize the corresponding sugar esters. Slight changes were brought to the present process and even though both synthesis methods are still very similar, those changes must be considered. First off, a different honey was used; in the first research article a standard European flower honey was taken whereas in the current work MH was chosen and used as a substrate. Second off, the novelty that was brought herein is the use of vinyl myristate (C14) instead of the palmitate vinyl ester (C16). MH was chosen as a substrate and solvent for carrying this reaction due to its well-established bioactivity that is higher than other honeys and displays efficacy against several pathogens [31,32,33]. The aim herein was first to select the honey presenting the best bioactivity according to literature and second to enhance its bioactivity with in situ synthetized glycolipids displaying several aliphatic chain lengths. It was expected that bringing exogenous surfactants into MH, which intrinsically presents a versatile pool of bioactive compounds, would make a real difference in terms of bactericidal effect. This endogenous pool of compounds that are present in MH is mainly composed of MGO, bee defensin-1, and a vast range of flavonoids as well as phenolics [34].

The approach that was used for this work was first reported by Pöhnlein et al., then by Delavault et al. and corresponds to a strategy of a “two-in-one” system in which sugars are part of the solvent and part of the substrates as they participate to the enzymatic conversion [11,35,36]. This methodology which was applied first to deep eutectic solvents (DESs) was successfully transferred to honey and agave syrup with chain lengths of esters ranging from C8 to C16 [12]. In the GOH mixture, it can be assumed that fructose and glucose laurate were produced and are responsible for the effect that will be discussed further in the next section. According to preliminary results, the C16 ester-enriched honey mixtures were not included in the present work but rather myristate (C14) was chosen alongside C8, C10, and C12 vinyl esters substrates for the lipase-catalyzed transesterification reaction. The immobilized lipase that was used is the versatile Novozym 435^®^ and it was expected that the reaction using vinyl ester with longer chains, notably C14, would be catalyzed in MH as it is clearly shown on the TLC analysis. The enzyme showed great flexibility towards the range of substrates that it can accept and the type of media in which it can remain active. MH, possessing as well a low water content and a low water activity [37], can reverse the lipase activity and form ester bonds the same way it was described in previous work [38,39].

However, one question remains concerning the effect of the process on compounds that are naturally present in MH. Hydrogen peroxide has been showed to be the main factor for bioactivity in standard honey [40]. This last compound can be found in MH as well but the component being unique and mainly responsible for its activity is the MGO which originates from the conversion of dihydroxyacetone that is present naturally in Manuka nectar flower [41]. Both of these components are known to be temperature-sensitive and studies have shown that the storage of MH at 37 °C leads to an apparent non-enzymatic increase of MGO after several days [42]. The process that was employed here to enrich MH uses 50 °C for 48 h, so in theory, an increase of MGO should be observed and, therefore, an increase in bioactivity. Withal, MHWE which serves as a negative control since no glycolipids are present in this condition, shows no increase in bioactivity compared to pure MH even after heat treatment (Figure 3). Therefore, it can be affirmed that glycolipids are responsible for the observed increase in bioactivity. Still, composition in MGO and the related components after processing could be an additional avenue of investigation.

### 3.2. Role of the Glycolipids in Bioactivity Enhancement

Previous discussion made it obvious that glycolipids in honey are playing a role in the bioactivity of the mixture. Nonetheless, the susceptibility test, stress assay, and antibacterial test were applied in order to get a deeper understanding on how this class of bio-surfactants acts inside the honey mixtures and in what measure they enhance the overall bioactivity of MH.

The susceptibility test showed clear evidence of MH and glycolipid mixtures to affect different types of microorganisms. Albeit, the evidence that glycolipid in honey brings bioactivity enhancement is not very clear from this first experiment. The bacteria and yeast that were tested exhibit clear sensitivity to honey mixtures at full concentration. The high osmolarity and unique MGO compound among other factors of MH could explain this remarkable effect [43]. Nevertheless, the test with *Candida bombicola* suggests that enzymatic glycolipid synthesis in honey could bring another strong antimicrobial factor to this sugar-supersaturated solution. The different sensitivities that were displayed by those microbes guided us to investigate further the GOH mixture which seemed to be the most efficient out of all the glycolipid enriched mixtures. A clear tendency of Gram+ pathogens being more sensible to all the honey mixtures was highlighted then compared to the other tested microorganisms and visibly no further bioactivity enhancement against Gram+ species was brought to MH throughout this experiment. Bacteria are known to be intolerant to high osmolarity, thus explaining the overall effect on non-yeast species with or without glycolipids. Interestingly the low susceptibility for *P. putida* could be explained by the polysaccharide slime capsule that *Pseudomonas* species possess [44]. However, the yeast species and Gram– pathogens displayed high sensitivities to sugar ester-enriched honey. Wagh et al. already reported a high sensitivity of Gram+ bacteria compared to Gram– when exposed to sugar esters [45]. We can postulate that a similar effect that was observed in this work is due to Gram+ bacteria and yeasts only possessing a single lipid membrane while Gram– bacteria are equipped with an additional one. Therefore, honey that is enriched with glycolipids renders Gram– bacteria and yeasts more susceptible to the high osmolarity of honey but make almost no difference in bioactivity against Gram+ ones.

Yeasts, such as *C. bombicola*, normally tolerate high osmolarity [46], but in our case we observed an increased effect as the chain length of the hydrophobic tail is shortened such as C8 > C10 > C12, suggesting a higher toxicity of acyl derivates possessing short alkyl chain and no effect with pure unmodified MH. The toxicity of C8 fatty acid was previously reported against *Saccharomyces cerevisiae* showing a general tendency of yeasts to be sensitive to relatively short, saturated acid chains as it damages the membrane by inducing cell leakage [47]. Interestingly *Candida bombicola* is a yeast that is able to produce sophorolipids [48], which is made of a glucose-derived di-saccharide acylated with C16 or C18 fatty acid tails. Thus, giving a possible insight on why honey mixtures that are provided with long chain vinyl fatty acid (e.g., C14 and C16) have very little effect against *Candida bombicola* compared to the C8 sugar esters. It is also important to mention that the *Candida* genus is known to produce a plethora of lipases that are commercially available [49]. Then, another conjecture could be that a lipase from *C. bombicola* is indirectly responsible for the observed effects as glucose octanoate represents a substrate that releases C8 free fatty acid, which is fatal to the yeast when GOH was used as a stressor. However, this hypothesis is less likely as no effect with MHWE containing unreacted octanoate vinyl esters was observed.

Although in this study an apathogenic yeast was surveyed, *Candida* represents the genus of the most common cause of fungal infections with *C. albicans* as the most prominent species [50]. Alongside *C. glabatra*, *C. tropicalis*, *C. parapsilosis,* and *C. krusei*, these five species represent 90% of infection cases among the *Candida* genus [51]. Therefore, follow-up studies might focus on the potential of the GOH mixture as a fungistatic, since *C. albicans* is known to grow as biofilms on human tissue and implanted medical devices [52].

The stress assay combined with an antibacterial test allowed a quantitative insight on possible mechanisms that are involved into the microorganisms´ inhibition that is induced by MH and glycolipid-enriched MH mixtures. For this test, *E. coli* was chosen as a whole cell biosensor (RGB-S) because its expression can be measured. Reading of OD_600_ during the 24h incubation of the biosensor gave clear confirmation that GOH showed quicker and stronger inhibition potential than any other pure or glycolipid-enriched MH at only 1.6% (*v*/*v*) concentration. In addition to that, a remarkable 4-h lag phase can be observed at the beginning of the incubation time with the GOH condition. Only concentrations ranging from 0 to 1.6% (*v*/*v*) in broth dilution were tested because of phenolics that are contained in honey interfering with the readings. Any measurement above these concentrations were thus rendered inoperable. The mixture that was composed of octanoate esters of honey sugars revealed a 6-fold increase of the physiological stress and a 2.5-fold increase of the cytotoxic stress compared to MH while having no impact on the microbes’ genetic material. Three main clarifications may be evoked in attempt to explain these observations. First, sugar esters are known for their antibacterial activities because of their ability to disrupt membranes [53,54,55]. Second, it appeared within our experiments that shorter chain length of the sugar ester resulted in an increase of bioactivity. Indeed, shorter alkyl chains resulted in higher critical micelle concentration (CMC) which seems to explain a higher antibacterial potential [56]. The toxicity of glycolipids towards pathogens inside the honey mixtures appears to increase as the chain length decreases which appears logical as the latter can greatly modulate the physicochemical properties of the sugar esters [57,58]. Third, since the glycolipids seem to spare DNA damage inside the cell, it is less likely to induce mutations due to genotoxic stress thus making it a good candidate for a potential antibacterial treatment that would not induce antimicrobial resistance [59,60,61]. Finally, the antibacterial test highlighted better efficiency of GOH at 20% (*v*/*v*) as no inhibition was observed for MH upon this same concentration in an agar plate assay. As a correlation to a previous stress assay experiment, it can be confirmed that glycolipids bring a supplementary factor to the bioactivity of honey.

Despite these results and speculations, similar experiments with isolated glycolipids that are produced in MH should be carried in a further investigation to potentially highlight a synergy between honey and glycolipids. As honey and DESs are physicochemically related, a synergy can be indeed suggested as DES formulations containing bioactive compounds exhibit such synergistic behavior [62,63,64,65]. Giving those facts, we can hypothesize that glycolipid bioactivity is enhanced by honey and vice versa due to multiple physico-chemical factors inherent to both participants.

## 4. Materials and Methods

### 4.1. Materials

#### 4.1.1. Chemicals

Lipase B from *Candida antarctica* that was immobilized on acrylic resin (Novozym 435^®^), Mueller Hinton Broth, kanamycin and LB broth were purchased from Merck Chemicals GmbH (Darmstadt, Germany). Commercially available Manuka honey MGO 550+ (Manuka Health, Te Awamutu, New Zealand) was purchased from Real GmbH (Karlsruhe, Germany) and used as both substrate and reaction media. All fatty acid vinyl esters were acquired from Tokyo Chemical Industry Co., Ltd. (TCI-Europe, Zwijndrecht, Belgium). Ultrapure water that was used for dilution of the MH and honey mixtures was obtained with a Purelab flex purification system (Ransbach-Baumbach, Germany). European agar was obtained from Becton Dickinson (Le Pont de Claix, France). Rhamnolipid standard mixture JBR599 was purchased from Jeneil Biosurfactant Co. (Saukville, WI, USA). If not stated otherwise, all other chemicals were purchased from Carl Roth (Karlsruhe, Germany).

#### 4.1.2. Microorganisms, Construction and Assembly of the RGB-S Reporter

Microbicidal tests were assessed against methicillin-resistant *S. aureus* (MRSA DSM 11729), *B. subtilis* (ATCC 21332), *C. bombicola* (ATCC 22214), *E. coli* (K12 DSM 498), and *P. putida* (DSM 5235). The RGB-S reporter was constructed by fusion of three synthetic sensing elements, cloned into the backbone pMK-RQ [kanR & ColE1 ori] (GeneArt^®^ Gene Synthesis, ThermoFisher Scientific, Karlsruhe, Germany and IDT Inc., Coralville, IA, USA). Each sensing construct is comprised of three parts: (a) stress-responsive promoter, (b) a fluorescent reporter protein, and (c) a transcriptional terminator. Based on the main criteria of exhibiting a fast and specific response, stress-responsive promoters were selected based on the literature. Sequences of the chosen promoters were obtained from the genome sequence of *E. coli* str. K12 substr. MG1655 (GenBank: U00096.3). Fluorescent proteins were chosen with respect to high signal intensity and spectral compatibility. To enhance their translation rates, codons for the selected fluorescent proteins were optimized for *E. coli* using the GeneOptimizer™ software (ThermoFisher Scientific, Karlsruhe, Germany) [66]. The sequences of fluorescent proteins as well as transcriptional terminators were obtained from the well documented parts of the Registry of Standard Biological Parts (parts.igem.org). For the handling of genetic designs in silico, the software Geneious 9.1.8 (Biomatters Ltd., Auckland, New Zealand) was used. The RGB-S reporter was then assembled using the isothermal cloning reaction [67] Gibson Assembly^®^ Master Mix (NEB Inc., Ipswich, MA, USA) according to the manufacturer’s protocol. If necessary, the template DNA was removed by DpnI treatment. A total of 1 µL of methylation-sensitive restriction enzyme DpnI and 2 µL CutSmart^®^ buffer (both from NEB Inc., Ipswich, MA, USA) were added to the assembled reaction and incubated at 37 °C for 30 min. The reaction product was used to transform the chemical competent *E. coli* cloning strain DH5α (lab stock) then plated on LB+Kan agar and incubated overnight at 37 °C. All the plasmids were purified using the ZR Plasmid Miniprep-Classic (Zymo Research Inc., Freiburg im Breisgau, Germany) following the manufacturer’s protocol, sequence verified (LGC genomics), and stored at −20 °C. After initial construction of the dual-color sensor (named RG-S reporter) consisting of PsulA::GFPmut3b::terminator_1 and PosmY::mRFP1::terminator_2, this vector was linearized using primers O17051-F and O17052-R and assembled with the third sensing element (terminator_3::PgrpE::mTagBFP2::terminator_4), resulting in the final triple-color/stress sensing plasmid named RGB-S reporter.

### 4.2. Methods

#### 4.2.1. Preparation of Glycolipid-Enriched Manuka Honey

The enzymatic synthesis of glycolipids in Manuka honey (MH) is based on Siebenhaller et al. [12] with slight modifications as follows: 200 mg of Novozym 435^®^; 1.03 μmol of pure fatty acid vinyl ester (vinyl octanoate, vinyl decanoate, vinyl laurate, vinyl myristate and vinyl palmitate); and 2.5 mL of MH were filled in a 5 mL Eppendorf tube. After 30 s vortex homogenization, the reaction was carried out in a rotator mixer with U2 program at 50 rpm (neoLab, Heidelberg, Germany) at 50 °C for 48 h. A total of five different glycolipid-enriched MH mixtures (designated as stressors) were formed containing sugar esters of octanoate (GOH), decanoate (GDH), laurate (GLH), myristate (GMH), and palmitate (GPH). An additional mixture which did not contain enzyme but only 1.03 × 10^−3^ mmol of vinyl octanoate (MHWE) was also incorporated to the antibacterial tests as a comparison.

#### 4.2.2. Extraction and Detection of Glycolipids via Thin Layer Chromatography (TLC)

The synthesized glycolipids were extracted from the honey media by the addition of 2 mL of warm water and homogenization of the resulting mixture. After the addition of 3.5 mL ethyl acetate and vortexing for 30 s, a glycolipid-containing organic phase was formed and further used for TLC analysis as follows. A total of 10 μL of the previously extracted organic phase were spotted onto a silica plate (Alugram SIL G, 60 Å, Macherey-Nagel GmbH & Co. KG, Düren, Germany). The eluent consisted of chloroform: methanol: acetic acid (65:15:2 *v*/*v*) [11]. After elution, the TLC plate was dipped into anisaldehyde: sulfuric acid: acetic acid (0.5:1:100 *v*/*v*) dying solution and subsequently revealed with a heat gun.

#### 4.2.3. Susceptibility Test

The susceptibility test with the different microorganisms and glycolipid-enriched mixtures was qualitatively assessed using an agar well diffusion assay according to Mavric et al. [7]. The microorganisms were pre-cultivated overnight at 37 °C (MRSA, *E. coli* and *P. putida*) or 30 °C (MRSA, *B. subtilis* and *C. bombicola*) in 100 mL flasks containing 10 mL of nutrient broth according to [68]. Afterwards, 100 μL of the undiluted cultures (OD_600_ = 0.4) were spread on plates containing solidified nutrient medium. The wells that were 8 mm in diameter (150 μL capacity) were punctured into the surface of the agar medium. A total of 120 μL of solutions of the enriched honey mixtures, MH or MH plus fatty acid containing no enzyme were placed into the wells. The plates were incubated at 37 °C or 30 °C depending on the microorganisms and observation on the inhibition zone was made after 24 h.

#### 4.2.4. Stress Assay Protocol

For the stress assay, an aliquot of the seedbank was used to inoculate two independent cultures (Cult. 1 & Cult. 2) each in 5 mL LB+kan broth and incubated overnight at 180 rpm and 37 °C. The next day, the cultures were diluted to 1:250 using fresh LB+kan and incubated again at the same conditions for 5–6 h. After reaching an adequate optical density (OD_600_ = 0.8), the cultures were diluted using fresh LB+kan to adjust the OD_600_ to 0.4, which is 2-times the final cell density in the assay plate. The honey mixtures were diluted as well in LB+kan to 2-times the final required concentration. The stress treatment started by adding 250 µL diluted sensor strain culture to 250 µL LB-stress mixture (or LB containing no honey mixtures as the control culture) to form 500 µL total volume, which had a final cell density of OD_600_ = 0.2 and a fixed stressor concentration (honey mixtures). Upon mixing, each of the two cultures (Cult. 1 & Cult. 2) were then distributed in triplicates in 96-microwell plate, each well containing 150 µL. In total, six independent biological replicates were analyzed for every honey mixture concentration unless indicated otherwise. The tested concentrations of MH, MHWE, GOH, GDH, GLH, and GMH were 0, 0.1, 0.2, 0.4, 0.8, 1.2, and 1.6% (*v*/*v*), respectively. The microtiter plate was covered by a fluorescence-compatible transparent film (Lab Logistics Group Inc., Meckenheim, Germany) to prevent culture evaporation and incubated in a Synergy H1 microplate reader (BioTek Inc., Bad Friedrichshall, Germany) with continuous orbital shaking (282 rpm, 3 mm) at 37 °C and measuring the optical density (OD_600_) and the three fluorescence readouts corresponding to the different types of stress. For the fluorescence readouts, respectively, the excitation and emission peaks correspond to the following wavelengths: red fluorescent protein (RFP, physiological stress): 571 nm, 607 nm; blue fluorescent protein (BFP, cytotoxic stress): 400 nm, 454 nm; and green fluorescent protein (GFP, genotoxic stress): 483 nm, 511 nm.

#### 4.2.5. Assessment of Antibacterial Activity

The antibacterial activity of GOH and MH was quantitatively assessed using an agar well diffusion assay according to Patton et al. [69]. *E. coli* was pre-cultivated overnight at 37 °C in 100 mL flasks containing 10 mL of Mueller Hinton Broth (MHB) medium. Afterwards, 100 μL of the cultures (OD_600_ = 0.1) were spread onto plates containing solidified MHB agar. Wells that were 15 mm in diameter (200 μL capacity) were punched in the broth-agar medium. A total of 150 μL of solutions of the GOH and MH that were diluted to concentrations ranging from 0% being the control well to 50% in ultrapure water were placed into the wells. The plates were incubated at 37 °C for 24 h. The zones of inhibition were measured using pictures that were taken into a FAS Digi Imaging System (FastGene^®^, Nippon Genetics Europe GmbH, Düren, Germany) with a GF-7 digital camera (Panasonic Europe GmbH, Hambourg, Germany). The measurements were then made with ImageJ software (NIH, Bethesda, MD, USA).

## 5. Conclusions

It was shown that production of glycolipids in MH, structurally identified as fructose and glucose laurate in previous work, catalyzed by the lipase formulation Novozym 435^®^ enhances bioactivity of the mixture.

The stress and antibacterial assay revealed bioactivity strengthening even upon high dilution factors. The mixture that was dubbed GOH, enriched with octanoate sugar esters induced 6-fold physiological stress increase and 2.5-fold cytotoxic stress increase compared to normal MH alongside a stronger and quicker inhibition based on the OD_600_ readings. On the other hand, no significant genotoxic stress signal from *E. coli* was measured. Such glycolipid-enriched honey mixture represents a potential tool to rapidly and efficiently treat wounds that are susceptible to getting infected. In a practical case, they could be administered “as is” in a pasture-like form with minimal prior downstream processing steps. Nonetheless, more investigation regarding the effect of the isolated glycolipids that were produced herein should be carried out to highlight a suspected synergistic effect. As an outlook, liquid chromatography using light scattering analysis coupled with mass spectrometry could help to improve the quantification and the characterization of the compounds that are present in our mixtures.

## Figures and Tables

**Figure 1 ijms-23-12031-f001:**
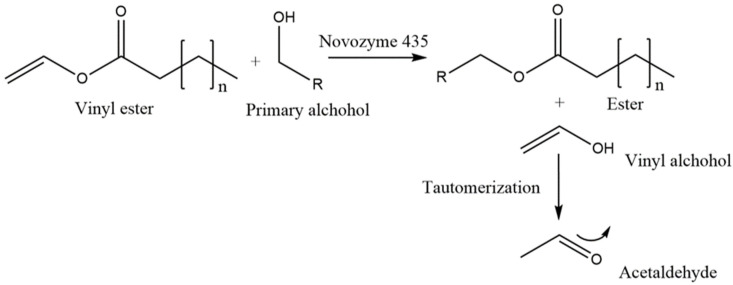
Immobilized lipase catalyzed transesterification between primary alcohol from honey sugars (R = glucose, fructose) and vinyl esters leads to newly formed esters (n = 6, 7, 9, 11, 13) thus producing enriched Manuka honey mixtures that are dubbed as GOH, GDH, GLH, and GMH, respectively.

**Figure 2 ijms-23-12031-f002:**
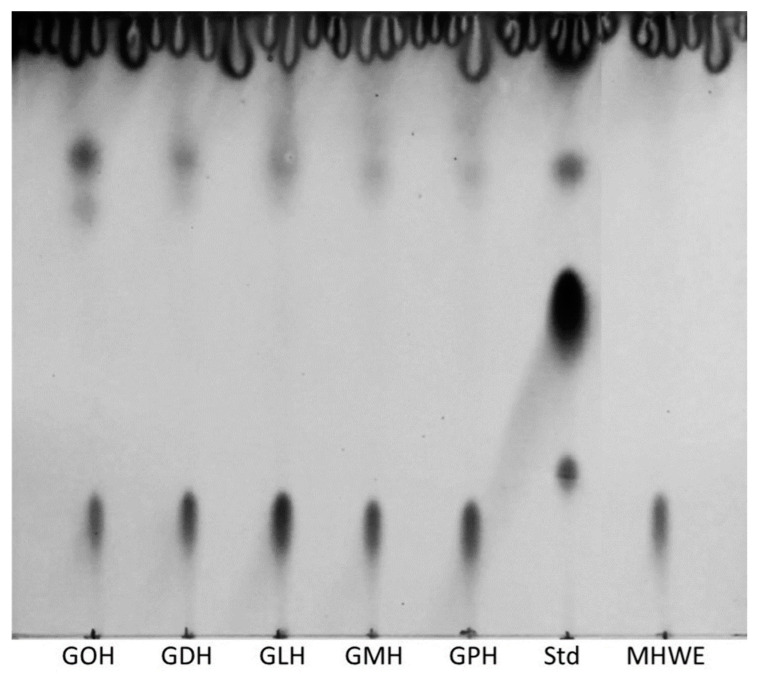
Visualization of the synthesized glycolipids in Manuka honey after dying with an anisaldehyde solution. A total of 10 μL of extracted honey mixtures and 4 μL of the standard were spotted on the TLC plate. Std, intern laboratory rhamnolipid standard; GOH, glycolipid octanoate honey; GLH, glycolipid decanoate honey; GLH, glycolipid laurate honey; GMH, glycolipid myristate honey; GPH, glycolipid palmitate honey; MHWE, Manuka honey without enzyme.

**Figure 3 ijms-23-12031-f003:**
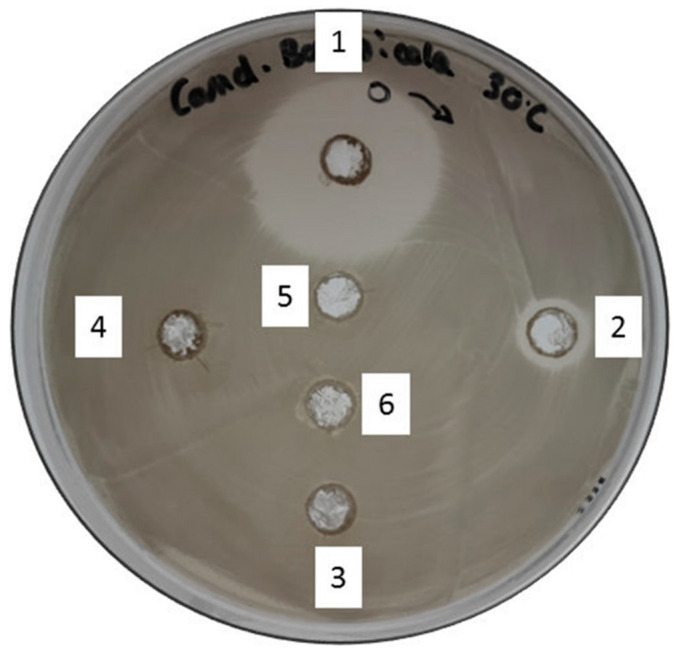
Qualitative agar well diffusion assay (*Candida bombicola*). Every honey mixture was incorporated undiluted in the wells. Well 1, GOH; Well 2, GDH; Well 3, GLH; Well 4, GMH; Well 5, Manuka honey; Well 6, Manuka honey mixture with 0.31 mmol/mL vinyl octanoate and no enzyme (MHWE).

**Figure 4 ijms-23-12031-f004:**
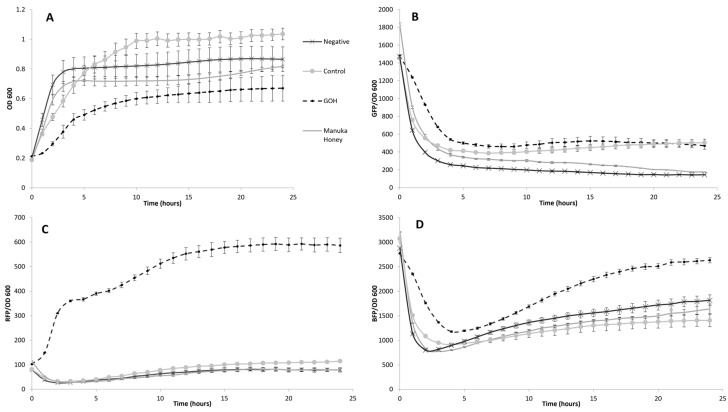
Biosensor-based stress assay using the designed *E. coli* strain. The bioactivity and the type of stress that was induced by the different honey mixtures were assessed with a multi-stress whole-cell biosensor set-up called RGB-S reporter. The readout displays: (**A**) OD_600_ (culture growth). (**B**) GFP (genotoxic stress). (**C**) RFP (physiological stress). (**D**) BFP (cytotoxic stress). Signals are normalized to the OD_600_ of the biosensor giving specific signals for each type of stress. 24 h kinetic with one measure per hour at 1.6% (*v*/*v*) displaying bioactivity of MH, MHWE and GOH. Measurements were done in triplicate using two different cell cultures (n = 6).

**Figure 5 ijms-23-12031-f005:**
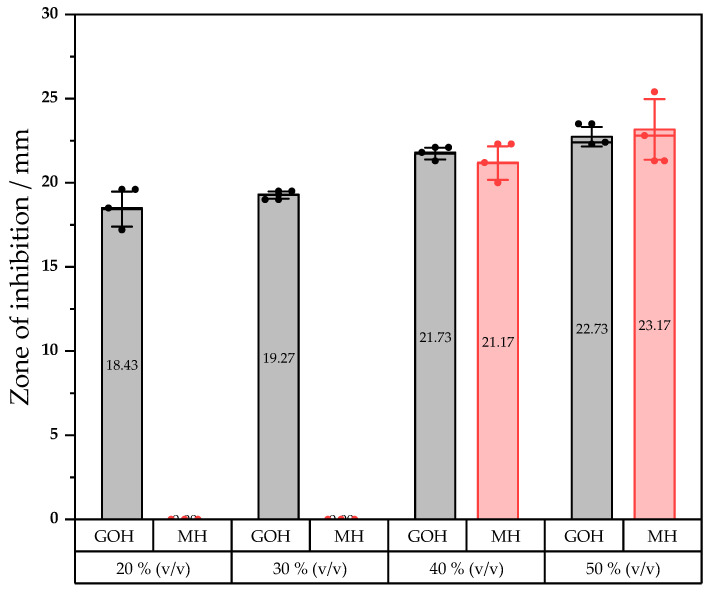
Zone of inhibition that was introduced by GOH and MH against *E. coli* at various concentrations. The wells were done in triplicate across different agar plates (n = 3).

**Table 1 ijms-23-12031-t001:** Qualitative agar well diffusion assay highlighting microorganism susceptibility allowing a relative comparison between mixture efficacies.

Microorganisms/Cell Type	MH	MHWE	GOH	GDH	GLH	GMH
MRSA/Gram+	++	++	++	++	+	+
*B. subtilis*/Gram+	+++	+++	+++	+++	+++	+++
*C. bombicola*/Yeast	0	0	+++	+	0	0
*E. coli*/Gram–	++	++	+++	+++	+++	+++
*P. putida*/Gram–	+	+	++	+	+	+

Note: Effect is categorized based on the diameters of the inhibition zones: +++, diameters ≥ 29 mm. ++, diameters < 29 mm and ≥19 mm. +, diameters < 19 mm and >0 mm. 0, no observable effect. MH, Manuka honey; MHWE, Manuka honey and unreacted vinyl octanoate (negative control for glycolipid synthesis: approach without enzyme); GOH, glycolipid octanoate honey; GLH, glycolipid decanoate honey; GLH, glycolipid laurate honey; GMH, glycolipid myristate honey (n = 3).

## Data Availability

Not applicable.

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
