# Peer review of "Enhanced Bioactivity of Tailor-Made Glycolipid Enriched Manuka Honey"

_ijms, 2022, doi:10.3390/ijms231912031_

Round 1
Reviewer 1 Report
This work is concise and clear. The results obtained are well justified, although in the discussion there is an abundant use of hypotheses.
Comments:
Lines 69-70: the explanation about the work of reference 11 is not clear.
Figure 4: some comment on differences in bioactivity between MH and MHWE.
A brief characterization of the glycolipids tested should be reported. For example, how many fatty acids are incorporated into sugars? GOH, GDH, GLH and GMH should have similar amounts (moles) of fatty acids in order to compare their effects.
Author Response
Dear Reviewer 1, thank you for your comments and the time invested in our work. Please find the PDF in attachement containing the point-by-point responses.

Reviewer 2 Report
The paper by Delavault et.al. entitled “Enhanced bioactivity of tailor-made glycolipid enriched Manuka honey” described the synthesis and biological properties of a mixture of long chain esters from glucose and fructose. The synthesis of the esters function has been done by a bio catalytical approach using a polymer supported CAL B enzyme.
The work could be published after major revison.
I suggest the authors to consider the following points:
1) They synthesize different mixture of esters strating from C16 (palmitate) to C8 (Octanoate), due by the fact that the most active mixture is the C8, I suggest to try also the corresponding C6 and C4 derivatives.
2) There is only TLC analysis to demonstrate the product, I suggest to try the purification of the mixture to do also HRMS and fully NMR analysis
3) The NMR analysis is required because authors supposed that the fructose and glucose has been esterified only in the primary alcohol moiety, but no demonstration of this finding are done.
4) I suggest the authors to do the synthesis of all esters starting from pure glucose and fructose to have in hands a standard compound useful for the biologicals testing and characterization.
5) Is not clear how they perform the stress assay, in particular the reference 20 is not a paper, seems to be a PhD dissertation (in german). To better understand the methodology, I suggest to add a methodology part in which the authors describe the E coli strain with the plasmid inserted and in particular what kind of promoters are responsible for the production of the different RGB-S reporter.
6) How does it means physiological stress? What kind of molecule or protein they are looking during this experiment? The same for the others type of stress induced fluorescence.
7) In the bibliography, ref. 17, 18 and 20 are incomplete.
Author Response
Dear Reviewer 2,
We are thankful for your valid and righful comments as well as for the time you invested in our work.
Please find attached as a PDF, our point-by-point responses.
